Long-read viral metagenomics captures abundant and microdiverse viral populations and their niche-defining genomic islands

Warwick-Dugdale Joanna 1 2
Solonenko Natalie 3
Moore Karen 2
Chittick Lauren 3
Gregory Ann C. 3
Allen Michael J. 1 2
Sullivan Matthew B. 3 4
Temperton Ben b.temperton@exeter.ac.uk 2
1 Plymouth Marine Laboratory , Plymouth , Devon , United Kingdom
2 School of Biosciences, University of Exeter , Exeter , Devon , United Kingdom
3 Department of Microbiology, Ohio State University , Columbus , OH , United States of America
4 Civil, Environmental and Geodetic Engineering, Ohio State University , Columbus , OH , United States of America
Nelson Craig
Electronic publication date: 2019 Apr 25
Publication date: 2019
Volume: 7
Electronic Location ID: e6800
Received 2018 Nov 15; Accepted 2019 Mar 14
Copyright: ©2019 Warwick-Dugdale et al.
Copyright year: 2019
Copyright holder: Warwick-Dugdale et al.
License: This is an open access article distributed under the terms of the Creative Commons Attribution License, which permits unrestricted use, distribution, reproduction and adaptation in any medium and for any purpose provided that it is properly attributed. For attribution, the original author(s), title, publication source (PeerJ) and either DOI or URL of the article must be cited.
License URL: https://creativecommons.org/licenses/by/4.0/

Keywords: Viral Metagenomics, Virus, Virome, Metagenome, Assembly, Viral ecology, Long-read sequencing, Marine Microbiology

Funding: Bermuda Institute of Ocean Sciences as part of the BIOS-SCOPE program Royal Society and the Natural Environment Research Council (NERC) NE/P008534/1 NE/R010935/1 NERC Great Western Four+ (GW4+) Doctoral Training Partnership PhD NE/L002434/1 Gordon and Betty Moore Foundation #3790 5488 Major support was provided by a fellowship to Ben Temperton from the Bermuda Institute of Ocean Sciences as part of the BIOS-SCOPE program; the Royal Society and the Natural Environment Research Council (NERC) (NE/P008534/1 and NE/R010935/1 to Ben Temperton). Additional support was from a NERC Great Western Four+ (GW4+) Doctoral Training Partnership PhD to Joanna Warwick-Dugdale (NE/L002434/1) and the Gordon and Betty Moore Foundation (awards #3790 and 5488) to Matthew B. Sullivan. There was no additional external funding received for this study. The funders had no role in study design, data collection and analysis, decision to publish, or preparation of the manuscript.

==============================
Marine viruses impact global biogeochemical cycles via their influence on host community structure and function, yet our understanding of viral ecology is constrained by limitations in host culturing and a lack of reference genomes and ‘universal’ gene markers to facilitate community surveys. Short-read viral metagenomic studies have provided clues to viral function and first estimates of global viral gene abundance and distribution, but their assemblies are confounded by populations with high levels of strain evenness and nucleotide diversity (microdiversity), limiting assembly of some of the most abundant viruses on Earth. Such features also challenge assembly across genomic islands containing niche-defining genes that drive ecological speciation. These populations and features may be successfully captured by single-virus genomics and fosmid-based approaches, at least in abundant taxa, but at considerable cost and technical expertise. Here we established a low-cost, low-input, high throughput alternative sequencing and informatics workflow to improve viral metagenomic assemblies using short-read and long-read technology. The ‘VirION’ (Viral, long-read metagenomics via MinION sequencing) approach was first validated using mock communities where it was found to be as relatively quantitative as short-read methods and provided significant improvements in recovery of viral genomes. We then then applied VirION to the first metagenome from a natural viral community from the Western English Channel. In comparison to a short-read only approach, VirION: (i) increased number and completeness of assembled viral genomes; (ii) captured abundant, highly microdiverse virus populations, and (iii) captured more and longer genomic islands. Together, these findings suggest that VirION provides a high throughput and cost-effective alternative to fosmid and single-virus genomic approaches to more comprehensively explore viral communities in nature.

Introduction

The marine bacterial communities that regulate global carbon biogeochemical cycles are themselves structured by selective, phage-mediated lysis (Weinbauer, 2004; Suttle, 2007). Bacteria co-evolve with their phages and exchange genetic information, and phages even ‘reprogram’ hosts during infection so as to channel host metabolism towards phage replication (Forterre, 2013; Hurwitz, Hallam & Sullivan, 2013; Hurwitz & U’Ren, 2016). Over the last decade, the convergence of high throughput sequencing and the use of universal taxonomic marker genes for bacteria have revolutionised our understanding of microbial ecology (Torsvik & Ovreaas, 2002; Treusch et al., 2009; Thompson et al., 2017). Problematically, however, viral ecologists lack parallel approaches. First, PCR-amplified marker genes are limited to a narrow subset of the viral community, and require degeneracies and amplification conditions that undermine the quantitative nature of the data (Sullivan, 2015). Second, while short-read viral metagenomics studies to date have provided clues to viral function (e.g., virally encoded, host-derived central metabolism genes, known as Auxiliary Metabolic Genes: AMGs) (Breitbart et al., 2007; Hurwitz, Hallam & Sullivan, 2013), and first estimates of global viral gene abundance and distribution (Brum et al., 2015; Roux et al., 2016a), they suffer from technical limitations. This is because short-read assemblies are composites of populations ‘features’ (Mizuno, Ghai & Rodriguez-Valera, 2014), with successful assembly a function of coverage and branch resolution in assembly graphs (Temperton & Giovannoni, 2012; Olson et al., 2017). This limits our ability to assemble viral populations where multiple strains are abundant and microdiverse (Roux et al., 2017), as well as genomic regions of high diversity, such as genomic islands (GIs), which, in microbes, often contain niche-defining genes that drive ecological speciation (Coleman et al., 2006). In these latter regions, assembly is impeded by low coverage and/or repeat regions at the boundaries (Mizuno, Ghai & Rodriguez-Valera, 2014; Ashton et al., 2015).

These are not just technical limitations—emerging data suggests that these obstacles alter our understanding of viral roles on important taxa and global carbon biogeochemistry. For example, the globally dominant members of the chemoheterotrophic order Pelagibacterales comprise up to 25% of all bacterioplankton and are major contributors in the conversion of marine dissolved organic matter back to atmospheric CO2 (Giovannoni, 2017). Their associated viruses dominate global oceans (Zhao et al., 2013; Martinez-Hernandez et al., 2018) and are likely to contribute significantly to carbon turnover in surface water by release of labile intracellular carbon during lysis (Suttle, 2005; Suttle, 2007). However, the genomes of viruses associated with Pelagibacterales contain numerous GIs and/or high microdiversity (Zhao et al., 2013; Martinez-Hernandez et al., 2018). Such features fragment genomes in short-read assemblies, which reduces representation following contig size-selection for downstream analyses (Martinez-Hernandez et al., 2017; Roux et al., 2017). Though single-virus genomics (Martinez-Hernandez et al., 2017) and fosmid-based approaches (Mizuno et al., 2013; Mizuno et al., 2016) can overcome such issues, these methods are technically challenging and costly to implement.

Alternatively, recent advances in long-read sequencing technology might be leveraged to better capture microdiverse viral populations and genomic islands. Such approaches can yield very long reads (>800 kbp) (Jain et al., 2015; Jain et al., 2018; Loman, Quick & Simpson, 2015), which would be long enough to capture complete genomes of double-stranded DNA bacteriophages (‘phages’) (10–617.5 kbp (Mahmoudabadi & Phillips, 2018)). At a minimum, such long reads could span genomic global- and local repeat regions, which tangle the De Bruijn Graph and fragment the assembly (Koren & Phillippy, 2015). Long reads may also overcome assembly challenges in regions of low coverage, to improve overall assembly of genomes from both cultured isolates (Wick et al., 2017) and metagenomics (Frank et al., 2016; Driscoll et al., 2017). It is also probable that long-read assemblies using overlap-layout-consensus would be less prone to microdiversity-associated fragmentation of genomes observed in De Bruijn Graph approaches (Martinez-Hernandez et al., 2017; Roux et al., 2017).

The challenge is that long-read technologies (both from PacBio and Oxford Nanopore) currently require large amounts of input DNA (micrograms; Jain et al., 2018, instead of nanograms commonly available from natural viral communities in seawater; Hurwitz et al., 2013). Furthermore, PacBio subreads and nanopore reads have high error rates (5–10%), with the former enriched in insertion errors and the latter enriched in insertion-deletion errors (Weirather et al., 2017). Indel errors shift the reading frame of the DNA sequence and confound gene-calling algorithms, artificially inflating the number of identified stop codons and producing shorter gene calls (Warr & Watson, 2019). This is a particular problem for viral metagenomics as the median length of genes in dsDNA phages is approximately half that of their bacterial hosts (408 bp vs 801 bp, respectively) (Brocchieri & Karlin, 2005; Mahmoudabadi & Phillips, 2018), and the vast majority of viral genes in both dsDNA viral isolates and viral metagenomes (>50% and up to 93%, respectively) have no known function (Hurwitz & Sullivan, 2013; Mahmoudabadi & Phillips, 2018), making it difficult to evaluate the quality of gene calls from metagenomic assemblies.

Here, we adapted a Long-Read Linker-Amplified Shotgun Library (LASL) approach for quantitative viral metagenomics (Duhaime et al., 2012) to obtain sufficient quantities of high-molecular weight DNA from nanograms of viral community dsDNA for sequencing using the MinION sequencer from Oxford Nanopore Technology. We then established a bioinformatic workflow to combine such long-read data with complimentary short-read sequencing data to maximise the advantages and minimise the weaknesses of both sequencing technologies. Following validation on mock viral communities, we applied our new approach to the first marine viral metagenome from the Western English Channel (WEC). Here, we present the first use of long-read sequencing technology for viral metagenomics and show that this novel approach provides significant benefits when combined with short-read metagenomics.

Materials & Methods

Construction of the mock viral community

A mock viral community comprised of six isolated and sequenced marine Caudovirales with genome sizes ranging from 38.5–129.4 kbp was produced as described previously (Roux et al., 2016b). Briefly, viruses were cultivated from host Pseudoalteromonas or Cellulophaga via plaque assay, collected into MSM buffer (0.45 M NaCl, 0.05 M Mg, 0.05 M Tris base, pH 7.6) and purified by 0.2 µm filtration followed by treatment with DNase I (100 U/mL for 2 hr at RT; terminated by the addition of 0.1 M EGTA and 0.1 M EDTA). Viral capsids were enumerated via epifluorescence microscopy (SYBR Gold; wet mount method) (Noble, 2001; Cunningham et al., 2015). 1.4 × 109 virus particles from each culture were pooled, and DNA extracted via the Wizard® DNA Clean-up System (Promega A7280). DNA was quantified via Qubit fluorometer (Life Technologies).

Construction of the Western English Channel viral metagenome

A total of 20 L of seawater was collected in rosette-mounted Niskin bottles at a depth of 5m from the Western Channel Observatory (WCO; http://www.westernchannelobservatory.org.uk/) coastal station ‘L4’ (50°15.0′N; 4°13.0′W) on the 28th September 2016. Seawater was transferred immediately to a clean collection bottle, and processed to remove the cellular fraction (within 4 h of collection) via sequential filtration through glass fibre (GF/D: pore size 2.7 µm) and polyethersulfone (pore size 0.22 µm) filters in a 142 mm polycarbonate rig, with peristaltic pump. Precipitation of viruses from filtrate (denoted as the viral fraction) and primary concentration of virus particles was conducted by iron chloride flocculation and collection on 1.0 µm polycarbonate filters (John et al., 2011); filters were stored in the dark at 4 °C. Viruses were resuspended in ascorbate-EDTA buffer (0.1 M EDTA, 0.2 M MgCl2, 0.2 M ascorbic acid, pH 6.1), and transferred to Amicon Ultra 100 kDa centrifugal filter units (Millipore UFC910024) (Hurwitz et al., 2013) that had been pre-treated with 1% bovine serum albumin buffer to minimise capsid-filter adhesion (Deng et al., 2014) and flushed with SM buffer (0.1 M NaCl; 0.05 M Tris–HCl; 0.008 M MgCl2). Following concentration to 500–600 µL, virus particles were washed with SM buffer (Bonilla et al., 2016) and purified with DNase I (100 U/mL; 2 h at RT) to remove unprotected DNA (i.e., encapsulated DNA); DNase I activity was terminated by the addition of 0.1 M EGTA and 0.1 M EDTA (Hurwitz et al., 2013). Viral DNA was extracted from concentrated and purified viral particles using the Wizard® DNA Clean-up System (Promega A7280), removing PCR inhibitors (John et al., 2011).

Library preparation, amplification and sequencing

For short-read sequencing, Illumina libraries were generated from 1 ng of either mock viral community DNA (Table S1), or 1 ng of environmental viral-fraction DNA, using Nextera XT v2 kits (Illumina) and the manufacturer’s protocol. After 12 cycles of amplification, the concentration and distribution in fragment sizes of the Illumina libraries were determined via Qubit and Bioanalyzer (Agilent), respectively. DNA was sequenced as 2 × 300 bp paired-end sequence reads, on a HiSeq 2500 (Illumina Inc.) in rapid mode, by the Exeter Sequencing Service (University of Exeter, UK).

For VirION libraries (Fig. 1), 20 ng (mock viral community) or 100 ng (WEC viral-fraction) of DNA was sheared to fragments averaging 8 kbp length via g-TUBE (Covaris 520079) as required to optimise MinION flow cell sequencing efficiency/yield (Oxford Nanopore Technologies: ONT). End-repair of DNA fragments, amplification of DNA with PCR-adapter ligation (i.e., Linker Amplified Shotgun Library: LASL preparation), and preparation of MinION-compatible libraries were performed following the manufacturer’s protocols for “2D Low input genomic DNA with PCR” using the ‘Ligation Sequencing kit 2D’ (ONT SQK-LSK208). PCR reaction conditions were modified with reference to NEBNext High-Fidelity 2X PCR Master Mix (NEB M0541S) manufacturer’s instructions in order to maximise DNA yield, whilst minimising production of chimeric sequences, as follows: 3 m at 95 °C (initial denaturation), 15 cycles of: 15 s at 95 °C (denaturation), 15 s at 62 °C (annealing), 5 min at 72 °C (extension); finally, 5 min at 72 °C (final extension), followed by 0.4 × AMPure bead clean-up. ∼1.5 µg of end-repaired, amplified DNA was carried forward for sequencing adapter ligation followed by purification of adapted DNA using MyOne C1 Streptavidin beads (Thermo Fisher Scientific Inc. 65001). The prepared long read library was sequenced on a single MinION Mk 1B flow cell with R9.4 pore chemistry for 48 h (Note—to remain up to date with changing ONT chemistry, a 1D ligation version of this protocol has also been tested and is available on protocols.io (https://www.protocols.io/view/virion-long-read-low-input-viral-metagenomic-sequ-p8fdrtn). Quality control of short and long read libraries was performed as described in Supplemental Information. Hiqh quality sequence data were used to generate short-read De Bruijn Graph assemblies (using metaSPAdes v. 3.11; Nurk et al., 2017), hybrid long-read scaffolded De Bruijn Graph assemblies (using metaSPAdes, with—nanopore parameter), and long-read overlap-layout consensus assemblies (with Canu; Koren et al., 2017) following optimisation for metagenomic data (see Supplemental Information). Rates of chimerism in both VirION reads (formed during PCR amplification) and assemblies (formed by mis-assembly) were evaluated by aligning reads and contigs, respectively, from the mock viral community to their associated genomes (Table S1).

Figure 1 Workflow for preparation of free-viral fraction DNA for MinION sequencing.

The long-read viral metagenomic method (VirION) developed includes FeCl3 flocculation and resuspension (FFR), shearing of extracted viral DNA (to 8–9 kbp), random linker amplification (Linker Amplified Shotgun Library: LASL), MinION library preparation, and nanopore (Oxford Nanopore Technologies; ONT) sequencing.

Maximizing the benefits of long read and short read assemblies

We developed a bioinformatic pipeline to maximise the benefits of VirION reads for viral metagenomics (Fig. 2). Briefly, long-read assembly contigs of VirION reads were ‘polished’ with matching short-read sequences to remove sequencing error via consensus base-calling (using Pilon; Walker et al., 2014, v1.22). In order to capture the longest assemblies available from the short-read data, scaffolds from short-read and hybrid assemblies were combined and dereplicated using a cut-off of 95% average nucleotide identity over 80% of the length (via MUMmer v3.23; Delcher, Salzberg & Phillippy, 2003) to cluster highly similar contigs into viral populations (Roux et al., 2016a). The longest representatives of each population were carried forward for analysis. Population representatives >10 kbp were pooled with polished long-read assembly contigs >10 kbp and evaluated with VirSorter (Roux et al., 2015) (in virome decontamination mode) to identify putative viral contigs. Reads classified as category 3 (deemed unusual, but not necessarily viral; Roux et al., 2015) were excluded from downstream analyses. Circular contigs (i.e., where the contig has matching ends) were identified by VirSorter and used as a proxy for successful assembly of a complete genome. Matching short-read data was then mapped against the representative viral population sequences (using bowtie2; Langmead & Salzberg, 2012) for use in evaluating (1) relative abundances of contigs; (2) whether long read assembly captured more microdiverse genomes; and (3) recovery of genomic islands and their predicted functional composition at the population level (see Supplemental Information).

Figure 2 Bioinformatics pipeline for VirION reads and complementary short-read sequencing for viral metagenomes.

The VirION bioinformatic pipeline to combine for short-read (Illumina) and long-read (MinION) sequencing to maximise the advantages of both sequencing platforms. Viral metagenomic short-read data and VirION reads from the Western English Channel were processed for identification of putative viral genomes as follows: (1) Short-read contigs and contigs scaffolded with VirION reads were generated via De Bruijn Graph Assembly using metaSPAdes (Nurk et al., 2017), and (2) de-replicated via average nucleotide identity of 95% similarity across 80% length. Separately, (3) long, error-prone VirION reads were assembled via overlap layout consensus Assembly using Canu (Koren et al., 2017) and (4) error-corrected via alignment of Illumina reads and consensus base calling with Pilon (Walker et al., 2014). (5) Putative viral genomes were identified using VirSorter (Roux et al., 2015). (6) Relative and global abundances of the Western English Channel viral contigs were calculated via competitive recruitment of short read data with FastVirome Explorer (Tithi et al., 2018), and lastly, (7) viral clusters based on shared proteins were produced from Western English Channel viral contigs clustered with contigs from the Global Ocean Virome (Roux et al., 2016a) and NCBI’s RefSeq database (v.8.4 among others—see Table S2) using vConTACT2 (Bolduc et al., 2017).

To direct future sampling efforts in environmental samples, we evaluated the short-read sequencing depth at which inclusion of long reads in hybrid assemblies offered no advantage in genome recovery. High-quality short read sequences were randomly subsampled in triplicate to seven discrete depths representing 10% and 70% of the full dataset (using seqtk https://github.com/lh3/seqtk). Subsampled reads were then assembled with and without scaffolding support from VirION reads (with metaSPAdes; Nurk et al., 2017). Scaffolds >10 kbp in replicated assemblies were classified as viral using VirSorter (Roux et al., 2015) in virome decontamination mode and the number of scaffolds classified as viral were calculated for each replicated assembly. Statistical significance of the number of viral or circular viral contigs between hybrid and short-read assemblies was calculated by a two-sided Student t-test between triplicate replicates at each sequencing depth.

Validation of error correction of long reads in viral metagenomic data

We evaluated whether it was possible to use short-read data to correct base-calling errors in long-read environmental metagenomic data in a similar way to that used for genomes of bacteria and eukaryotes from axenic samples (Walker et al., 2014). Western English Channel short-read data were sub-sampled to different sequencing depths, in triplicate; sub-samples were then mapped against the long-read assemblies of VirION reads. For error-correction with Pilon (Walker et al., 2014) and median coverage, the total number of fixed deletions and fixed insertions at each coverage depth were calculated. We then evaluated whether error-correction could be used to reduce the impact of frameshift errors on predicted gene length. Predicted coding sequences were identified using MetaGeneAnnotator (Noguchi, Taniguchi & Itoh, 2008) on the following: (1) uncorrected VirION reads; (2) long-read assemblies of VirION reads; (3) long-read assemblies of VirION reads polished with the full short-read dataset; (4) contigs from scaffolded short-read assemblies; (5) contigs from the hybrid assembly. Distributions of the lengths of predicted coding sequences were compared against those in the genomes of Caudovirales from the NCBI RefSeq database (v.8.4), predicted proteins from the Global Ocean Virome (GOV; Roux et al., 2016a), and the single-amplified viruses in Martinez-Hernandez et al. (2017). Effect size of different assembly types on genomic island length and density and associated 95% confidence intervals (CI) were calculated from bootstrapped medians (Cumming, 2014). For each bootstrap, 1000 predicted proteins were randomly subsampled from each dataset and their median length was calculated.

Analysis of Tig404—a contig closely related to Pelagiphage HTVC010P

Phage contigs closely related to Pelagiphage HTVC010P were identified via clustering of viruses at the ICTV-accepted level of genera by shared gene content (vContact2; Bolduc et al., 2017). Within this viral cluster, contig tig404, from the polished long-read assembly of VirION reads, was identified as a circular viral contig by VirSorter (Roux et al., 2015). Whole genome alignment was performed with MUmmer (Delcher, Salzberg & Phillippy, 2003) to calculate average nucleotide identity to HTVC010P. Contigs from short-read only and hybrid assemblies that shared 95% nucleotide identity over 80% of their length to tig404 were identified and mapped back to their respective loci with MUmmer (Delcher, Salzberg & Phillippy, 2003). Genomic islands and nucleotide diversity of tig404 were calculated as described previously. To evaluate the contents of a 5.3 kb genomic island, unpolished VirION reads were mapped back against the tig404 genome and those which mapped to at least 100 bp on the borders of the genomic island were extracted. Mapped reads extending at least 1 kb into the genomic island were used as a query in a tBLASTx best-BLAST (Camacho et al., 2009) search against the NCBI NR database to annotate the reads whilst minimising the adverse impact of sequencing error within the uncorrected reads.

Estimating relative abundance and viral clusters of WEC viruses in viral metagenomes

FastViromeExplorer (Tithi et al., 2018) v.1.1 was used to quantify the relative abundances of WEC viral contigs. FastViromeExplorer is built upon the Kallisto (Bray et al., 2016) framework and competitively recruits reads against contigs, allowing for accurate recruitment to contigs that may share a degree of sequence similarity. Briefly, high quality short read datasets from the Global Ocean Virome (Roux et al., 2016a) and from our Western English Channel sample were randomly subsampled to 10 million reads using seqtk (https://github.com/lh3/seqtk) to standardise per-sample sequencing effort, with the number of reads selected to balance detection of lower abundance viral populations with maximising the number of samples that could be included in the survey. Subsampled reads for each sample were recruited against a Kallisto index comprising (1) The viral genomes >10 kbp identified in this study; (2) A selection of phage genomes >10 kbp from key metagenomic studies (Roux et al., 2016a; Martinez-Hernandez et al., 2017; Luo et al., 2017); (3) Cultured viruses from the NCBI RefSeq viral database (v8.4) (Table S2). Contigs >10 kbp were selected to maximise accuracy of VirSorter to correctly identify viral contigs (Roux et al., 2015). For inclusion in downstream abundance analyses, contigs with less than 40% coverage as calculated by FastViromeExplorer were classified as having zero abundance to avoid over-representation of partial matches (Tithi et al., 2018). The top 100 most abundant contigs from each sample were also selected for downstream analyses. All phage genomes >10 kbp (including those from RefSeq) were processed using VirSorter (v.1.03) on the CyVerse cyberinfrastructure (Merchant et al., 2016) to standardise gene-calling prior to clustering of viruses into ICTV-recognised genera by shared gene content using vContact2 (Bolduc et al., 2017). In the final stage of clustering, vContact2 uses ClusterONE (Nepusz, Yu & Paccanaro, 2012) and assigns a p-value to a cluster depending on whether the in-cluster edge weights are significantly higher than the out-cluster edge weights. Q-values were calculated from cluster p-values using the qvalue R package (Dabney, Storey & Warnes, 2015) to account for multiple testing and a q-value cutoff of <0.05 was used to identify statistically significant clusters.

Results & Discussion

Here, we present the first use of long-read sequencing technology for viral metagenomics and show that this novel approach provides significant benefits when combined with short-read metagenomics. Our bioinformatics pipeline overcame the high sequencing error associated with long-read technology and the addition of long reads enabled capture of complete viral genomes which were globally ubiquitous, and not represented by short-read only assemblies. Long-read assemblies also significantly improved the capture of viral genomic islands, demonstrating that this advance will facilitate better understanding of niche-differentiation and ecological speciation of viruses in environmental samples.

Assembly of VirION reads successfully captured mock viral community genomes and retained relative abundance information

VirION sequencing of the mock viral community produced 359,338 high quality (Q>10) long reads (median length: 4,099 bp; max length 18,644 bp). 95% of the reads (341,718) mapped back to the genomes of the mock viral community. Considering viral DNA was sheared to 6–8 kbp fragments, the length of amplicons following LASL were shorter than expected, presumably due to preferential PCR amplification of shorter fragments (Shagin et al., 1999) (Fig. S1A) and preferential diffusion (and thus sequencing) of shorter reads within the flowcell microfluidics (Fig. S1B). Only 0.95% of LASL amplified reads were classified as chimeric (mapping to more than one location of the same or different genomes of the mock viral community), suggesting 15 rounds of PCR was sufficiently low to minimise production of chimeric artifacts, supporting previous findings (Laver et al., 2016). Several methods have been developed for sequencing dsDNA viral metagenomes without skewing relative abundance information important for comparative ecology, including an LASL approach optimised for 454 sequencing (Duhaime et al., 2012; Hurwitz et al., 2013) and Nextera sequencing (Roux et al., 2016b). Median per-genome coverages of VirION reads and short-read Nextera datasets (5.6 M 2 × 300 bp paired-end) from the mock viral community were strongly correlated (R2 = 0.975, p < 0.001, Fig. 3A), indicating that the LASL approach used here for multi-kilobasepair fragments retained relative abundance information observed in previous LASL approaches.

Long-read and short-read assemblies of the mock viral community captured >99.7% of the six mock viral community genomes (Table S1). Neither the short-read only assembly, hybrid assembly nor long-read assemblies were able to capture all six genomes in six complete contigs. Long-read methods gave the most contiguous assemblies, capturing the six genomes across 14 contigs. In comparison, short-read only assemblies recovered the genomes across 26 contigs, whereas hybrid assembly reduced the number of contigs to 21. As expected, we identified >250 times more indel errors in long-read only assemblies than in the short-read assemblies scaffolded with long reads (average of 474 vs <2 indels per 100 kbp, respectively). Polishing of long-read only assemblies with short read data reduced the indel error rate to 22.78 per 100 kbp, indicating this was a successful strategy for reducing indel error of long-read assemblies in metagenomic samples, but was not able to remove such errors completely. There was no evidence of chimerism in any of the assemblies, indicating that Canu’s in silico correction of chimeras (Koren et al., 2017) successfully removed the low number of chimeric sequences observed in the VirION reads during assembly.

Combining VirION reads with short read data improves viral metagenomic assembly in an environmental virome

Long read sequencing of an environmental virome from the Western English Channel produced 108,718 high quality VirION reads (median length: 3,625 bp; max length: 17,019 bp, total yield of 0.39 Gbp). It is worth noting that recent developments of MinION technology have improved flowcell yields to >10 Gbp (pers comms). Therefore, our analyses here represent low coverage of the viral community with long read data compared to currently available (and fast-improving) technology.

Scaffolding short-read assemblies using VirION reads provided a small, but significant increase in the number of putative viral genomes recovered (between 1.1 to 1.5-fold increase, Student t-test, p < 0.05) than short-read only assemblies up to a short-read sequencing depth of ∼12 Gbp (Fig. 3B). Above this depth, there was no significant difference between short-read assemblies with and without scaffolding, suggesting assembly of short-read data was capturing most of the viral community above this sequencing depth. For comparison, the median sequencing depth of 137 Illumina sequenced viral metagenomes from the Global Ocean Virome survey (study PRJEB27181 in the European Nucleotide Archive) was 8.67 Gbp (IQR = 5.22 Gbp), with 110 out of 137 samples sequenced to a depth of <12 Gbp. Inclusion of VirION reads in hybrid assemblies significantly increased the number of ‘complete’ (i.e., circular contigs) viral genomes recovered once short-read sequencing depth increased above 12 Gbp (1.5 to 2.0-fold, Student t-test, p < 0.05) (Fig. 3B). Details of differences in means and p-values at each depth are available in Tables S3 and S4. When the full (30.8 Gbp) short-read dataset was used, the inclusion of long reads for scaffolding De Bruijn Graph assemblies increased the median length of recovered viral genomes by an average of 1.8 kbp compared to short-read only assemblies (Mann–Whitney U test, n 1 = 1,400, n 2 = 879, p-value < 0.001). With an estimated mean gene density of 1.4 genes per kb in phage dsDNA genomes (Mahmoudabadi & Phillips, 2018), this increased length represents an extra 2.5 genes per contig.

Figure 3 Comparative performances of short-read and long-read data for the identification of marine viral genomes.

(A) Relative abundances of genome-mapped VirION reads and short-reads from a mock viral community composed of 6 different tailed bacteriophages. CBA: Cellulophaga phage; PSA: Pseudoalteromonas phage. The relative abundances of mock viral community members were strongly correlated using both approaches, showing amplification of sheared viral DNA for VirION sequencing was as quantitative as short read approaches for estimating relative viral abundance. (B) Efficiency of short-read only and hybrid sequencing approaches for detection of viral genomes at various depths/coverages of Illumina data using triplicate random subsamples of short read data from the Western English Channel viral metagenome: At all coverage depths tested, hybrid assemblies generated more circular (i.e., putatively complete) viral genomes than short-read assemblies; Below 10 Gbp of short-read data, hybrid assemblies captured more viral genomes (>10 kbp) than short-read assemblies. Comparisons within grey boxes were found to be statistically significant (Student t-test).

Polishing of long-read assemblies of WEC VirION reads using complementary short-read data removed a maximum of 172,854 insertion errors and 12,674 deletion errors (Fig. S2). Error correction reached an asymptote at ∼9 Gbp of short-read sequencing data, with a median coverage of ∼70. As expected, the errors associated with long-read sequencing adversely affected the lengths of protein predictions (Fig. S3), in accordance with previous findings (Warr & Watson, 2019). Proteins predicted from uncorrected VirION reads (median length of 72 aa, 70–74 aa 95% CI) were shorter (median difference = 61 aa, 69–53 aa 95% CI) than those from RefSeq Caudovirales genomes (median length of 133 aa, 126–141 aa 95% CI), and much shorter (median difference = 88 aa, 83–95 95% CI) than those from the GOV dataset (median length of 160 aa, 149–173 95% CI). Assembly of long reads with Canu includes a consensus-based error-correction step (Koren et al., 2017), which increased median predicted protein lengths to 87 aa (median difference of 15 aa, 14–15 95%CI) compared to raw VirION reads. Polishing of long-read assemblies of VirION reads with short read data was highly effective in restoring the length of predicted proteins (median length 127 aa, 120–135 aa 95%CI) to lengths similar to those observed in RefSeq Caudovirales (median length = 133 aa, 126–141 aa 95%CI). Proteins from polished reads had a median difference of −6 aa (−18–6 95%CI) compared to RefSeq Caudovirales proteins. This suggests that not all frameshift errors were corrected in the long-read assemblies, corroborated by evidence of increased indel errors observed in long-read assemblies of mock viral community data compared to short-read assemblies.

Interestingly, predicted protein lengths from the GOV dataset (Roux et al., 2016a) (median length = 160 aa), short-read only assembly of the WEC virome (median length = 157 aa); hybrid assembly of the WEC virome (median length = 160 aa) and data from single-amplified viral genomes (Martinez-Hernandez et al., 2017) (median length = 152 aa) were all of similar length and 19 to 27 aa longer compared to those from RefSeq Caudovirales genomes, and 25 to 33 aa longer than those from WEC polished long-read assemblies. In comparison, median predicted protein length in 899 dsDNA phages was previously estimated at 136 aa (Mahmoudabadi & Phillips, 2018)—similar to those found in our polished long-read assemblies from VirION reads. Thus, either both the RefSeq Caudovirales dataset and that of Mahmoudabadi and Phillips are under-representing longer viral predicted proteins found in marine viral metagenomes, or predicted protein lengths in viral genomes from metagenomic data are longer than those observed in cultured representatives. Whether this difference is biological or an artifact of metagenomic assembly and gene calling is an interesting area for further investigation.

Assembly and mapping of VirION reads captures more information about potential niche-defining genomic islands than short-read only or hybrid assemblies: In marine bacteria, genomic islands have been identified as playing an important role in niche specialisation that drives ecological speciation (Coleman et al., 2006). Genomic islands have also been found to be a common feature of viral genomes and are typically enriched in functions associated with host recognition (Mizuno, Ghai & Rodriguez-Valera, 2014). At all nucleotide identity cut-offs tested, genomic islands captured on long-read assemblies were between 145 bp (112–184 bp 95%CI) and 225 bp (189–259 bp, 95% CI) longer than those captured on short-read only or hybrid assemblies. (Fig. 4A, Fig. S4A). There were no significant differences between the lengths of genomic islands captured on short-read only or hybrid assemblies. The largest genomic islands in each assembly type were 2.47 kbp, 5.75 kbp and 5.65 kbp in short-read only assemblies, hybrid assemblies and long-read assemblies, respectively. In comparison, the largest genomic islands identified in fosmid-based viral metagenomes were ∼4.6 kbp (Mizuno, Ghai & Rodriguez-Valera, 2014), suggesting that both hybrid and long-read approaches capture similar length genomic islands as previous fosmid-based methods. Similarly, the density of GIs was significantly greater in long-read assemblies (at between 40 bp (20–60 bp, 95%CI) and 100 bp (80–110 bp, 95%CI) of GI per kbp of genome) compared to short-read or hybrid assemblies (Fig. 4B, Fig. S4B). Again, there was no significant difference between short-read only and hybrid assemblies. At a nucleotide identity cut-off of 98% for read mapping, the length of GIs in long-read assemblies were longer than those at 92% and 95%, (59 bp (18–106 bp, 95%CI) and 61 bp (13–105 bp, 95%CI) respectively), indicating that residual error in the polished reads may be contributing to a slight increase in predicted GI length and density at high nucleotide identity. However, these effect sizes are much smaller than those observed between long-read assemblies and short and hybrid assemblies across all identity cut-offs, suggesting that long reads do indeed improve the capture of genomic islands.

Figure 4 Long-read assemblies capture longer genomic islands than short-read methods.

Comparison of the (A) length of genomic islands (GI) and (B) normalised length of GI per kb of genome per contig captured on long read assemblies of VirION reads compared to short- read only and hybrid assemblies of viral contigs from the Western English Channel. Genomic islands were identified by mapping reads back against contig across a range of nucleotide percentage identities (92, 95, 98%) to account for residual error remaining in polished long- read assemblies. (C) and (D) represent pairwise significance calculated using a Wilcoxon Rank Sum Test, with p-values adjusted (Benjamini-Hochberg) for multiple testing, for (A) and (B), respectively. Effect sizes and 95% confidence intervals can be found in Fig. S4.

Previous work to identify viral genomic islands by recruiting short-reads back against assembled contigs from fosmid libraries showed that nucleotide diversity within genomic islands was associated with a constant-diversity model, with under-recruiting islands containing proteins associated with host recognition and penetration, phage structure and DNA packaging structural proteins (Mizuno, Ghai & Rodriguez-Valera, 2014). Here, we were able to gain further insight into viral genomic islands by investigating whether diversity also occurred at the functional level. VirION reads that spanned the full width of GIs were identified and their gene content was predicted at the nucleotide level. Reads spanning the same GI were compared to see if different proteins were encoded on different template strands prior to amplification.

In total, 137 genomic islands on 84 viral contigs had at least 10 VirION reads spanning their full length. The 3,072 reads spanning these islands encoded 6,445 predicted proteins, of which 4,599 could be aligned to a protein within the NR database. Just 711 (15%) of aligned predicted proteins returned a hit with known function, indicating that genomic islands are an important source of ‘genetic dark matter’ (i.e., sequence of unknown function) in viral metagenomes (Krishnamurthy & Wang, 2017). In total 66 genomic islands contained genes with an assigned function. These islands captured a range of functional proteins including those associated with nucleotide biosynthesis; DNA methylation; redirection of host machinery; structural proteins and associated chaperonins; endo and exonucleases and integrases (Table S5). 35 of the genomic islands contained structural proteins (capsid, tail proteins, co-chaperonin GroES, YapH); proteins associated with membrane recognition (carbohydrate-binding module, lectin-binding proteins) or proteins associated with reconfiguring host metabolic machinery for viral synthesis or defence suppression (RNA polymerase sigma factor, methyltransferases, tRNA synthetases, anti-restriction protein), supporting the hypothesis that viral genomic islands are a hotspot for Constant-Diversity evolutionary dynamics (Mizuno, Ghai & Rodriguez-Valera, 2014). Seven out of eight genomic islands containing a thymidylate synthase (an enzyme involved in pyrimidine metabolism) also encoded partial hits to ribonucleotide reductase (involved in both purine and pyrimidine metabolism). Ribonucleotide reductase has previously been identified as the nucleotide metabolism gene most frequently interrupted by self-splicing introns (Dwivedi et al., 2013). Similarly, thymidylate synthase has been found to contain self-splicing, group I introns in phage genomes (Chu et al., 1984; Bechhofer, Hue & Shub, 1994), potentially identifying intron splicing activity as a source of regulatory/functional variability and/or as a mechanism to promote the movement of genetic material within the viral genomic islands in our data. Functional, putatively niche-defining metabolic genes were also identified in the genomic islands, including an ultraviolet light damage repair gene uvsE and genes associated with photosystem II (psbA). Twenty-five out of the 66 genomic islands showed evidence of alternative gene arrangements across their spanning reads, suggesting the content of genomic islands can vary within viral populations at the structural, functional and nucleotide level.

Assembly of VirION reads capture important, microdiverse populations previously missed by short-read data

It has been hypothesised that genomes assembled from short-read metagenomes may be biased away from microdiverse populations (Martinez-Hernandez et al., 2017; Roux et al., 2017). We reasoned that overlap layout consensus assembly of long reads, followed by error correction might better capture genomes with high levels of microdiversity by avoiding the unresolvable branches of De Bruijn Graph assemblies. We evaluated genome-level nucleotide diversity (π) (Nei & Li, 1979) of both short-read assemblies and polished long-read assemblies from the Western English Channel virome. Median levels of π were significantly (3-fold) higher in polished long-read contigs than those derived from De Bruijn Graph assemblies (two-sided Mann–Whitney U test: W = 105,830, n 1 = 758, n 2 = 206, p = 4.81 × 10−15; Fig. S5), consistent with the hypothesis that long-read assembly of VirION reads captured genomes previously lost due to failure to resolve assembly graphs as a consequence of microdiversity.

Tig404—an example of how VirION reads improve viral metagenomics

The benefit of using VirION reads for viral metagenomics is exemplified by a polished contig from long-read assemblies that showed high nucleotide similarity and shared gene content to the globally abundant pelagiphage HTVC010P (Zhao et al., 2013). This ecologically important virus and its closely associated phages contain numerous genomic islands that comprise ∼10% of their genome and a shared 5.3 kbp genomic island containing a putative ribonuclease, bounded by tail fibre proteins (Mizuno, Ghai & Rodriguez-Valera, 2014). It has also been predicted to possess high microdiversity that challenges assembly from short-read data, leading to fragmentation and thus under-representation in short-read viral metagenomes, but is successfully captured using fosmid approaches and single-virus genomics (Mizuno et al., 2013; Martinez-Hernandez et al., 2017). Clustering of viral contigs from the WEC by shared-gene content using vContact2 (Bolduc et al., 2017) identified a virus called ‘tig404’ from long-read assembly of VirION reads that was 89% identical at the nucleotide level to HTVC010P. We mapped contigs from short-read only and hybrid assemblies against this genome at 95% nucleotide identity over 80% of the length to evaluate the success of short-read and hybrid assembly methods at capturing this genome, and identified its genomic islands as described above (Fig. 5). Both short-read only and hybrid assemblies were highly fragmented across the genome. Analysis of median nucleotide diversity of tig404 was extremely high (Fig. S5) and provided supporting evidence that fragmentation may be a result of high microdiversity in this phage. In contrast, VirION reads successfully overlapped across the genome and enabled recovery of the genome through long-read assembly. Comparison of the genome of tig404 with that of HTVC010P identified a shared genomic island containing a putative ribonuclease protein and bounded by a tail fibre protein (Fig. 5), similar to those observed in closely related taxa from fosmid libraries (Mizuno, Ghai & Rodriguez-Valera, 2014).

Figure 5 Long-read sequencing resolves microdiversity and assembly issues across genomic islands in ecologically important viral taxa.

De Bruijn Graph (DBG) assembly of short reads, even with VirION reads for scaffolding failed to assemble the genome of tig404, a virus closely related to the globally abundant pelagiphage HTVC010P. Only long-read assembly of VirION reads, followed by error correction with short read data was able to capture the complete genome on a single 29.2 kbp contig. A 200 bp sliding window analysis was used to calculate median coverage (A) of the assembly and (B) maximum nucleotide diversity (π), revealing six genomic islands (GIs) (C) and high levels of nucleotide diversity. The impact of this on short-read (light brown) only and hybrid assembly (green) can be seen in (C), where the assemblies aligned to the long-read assembly are highly fragmented. Conversely, long VirION reads (dark brown) were capable of spanning these regions across the whole genome and thus enabling assembly (D). One genomic island on tig404 was conserved with that of HTVC010P (E). Thus, we were able to identify the genomic content of this island at the population level by mapping VirION reads to HTVC010P and identifying those that spanned the genomic island. Encoded function was then predicted using tBLASTx to overcome high sequencing error in uncorrected VirION reads.

In addition, we were able to exploit an additional benefit of long reads and use unpolished VirION reads to explore the contents of the shared genomic island across the tig404 population within the WEC virome. As each read is derived from a single DNA strand (excluding the low abundance of chimeric reads), variance in the content of the genomic island within a population would be captured on reads that align to the ends, or across, the genomic island. In total, 31 VirION reads extended from the boundaries into the genomic island (Fig. 5). Of these, 17 had sufficient overlap to use for identifying functional genes. Those at the 5′ end of the genomic island all contained a putative ribonuclease, whilst those at the 3′ end all contained an internal virion protein thought to be associated with puncturing the cell membrane in T7-like phages (Mizuno, Ghai & Rodriguez-Valera, 2014). Thus, it would appear that, for this shared genomic island at the population level, diversity occurs at the nucleotide level, rather than gene content level. The fact that a similar gene content has now been found in the Western English Channel (this study), the Sargasso Sea (Zhao et al., 2013) and the Mediterranean (Mizuno, Ghai & Rodriguez-Valera, 2014) may indicate this is a conserved feature across the HTVC010P-like phages. The encoding of a ribonuclease within a genomic island offers an interesting glimpse into the host-virus interactions that occur during infection and suggests that degradation of RNA is an important feature of the arms-race in HTVC010P-like phages with their Pelagibacter hosts. Whether this is to shut down host metabolism, or to hijack host metabolism through manipulation of regulatory machinery enriched in riboswitches (Meyer et al., 2009) requires further investigation.

In total, analysis of VirION reads using a strategy to combine short and long read assemblies (Fig. 2) generated 2,645 putative viral contigs >10 kbp from the Western English Channel. Of these, 2,279 were from the de-replicated short and hybrid De Bruijn Graph assemblies and 366 from polished long-read assemblies. Our dataset represents the first virome sequenced from the WEC and so we evaluated the global abundance of viral populations from the WEC by competitive mapping of 10 million subsampled short reads from both the WEC and the GOV dataset (Roux et al., 2016a). Representatives of viral populations from the WEC were then pooled with those >10 kb from the GOV dataset and other marine virome datasets (Table S2) to make a total dataset of 20,545 viral contigs. Following competitive read recruitment with FastViromeExplorer (Tithi et al., 2018), the top 50 most abundant viral genomes were identified in each of the WEC and GOV surface samples. Out of 1,598 contigs, 81 of the most abundant viral contigs were from long read assemblies of VirION reads from the WEC, representing a significant enrichment (hypergeometric test for enrichment, p = 6.6 ×10−19). WEC contigs from short-read only (42 contigs) and hybrid assemblies (77 contigs) were not significantly enriched in the most abundant viral contigs. Thus, it is likely that long-read assembly of VirION reads from the WEC captured important and globally abundant viral taxa previously missed in the GOV datasets. Examination of relative abundance of WEC contigs in surface water samples from the GOV showed that contigs from long-read assemblies of VirION reads recruited a large proportion of the recruited reads from global samples, particularly in the Southern Atlantic Ocean and waters off the Western coasts of Southern Africa and South America (Fig. 6). In total, clustering VirION-derived contigs from the Western English Channel with contigs from previous studies (Table S2) by shared protein content produced 668 statistically supported viral clusters. Of these, 202 contained contigs derived from long-read assembly of VirION reads, but just three of these were comprised solely of these contigs. Thus, we are confident that previous findings suggesting viral diversity at the genera level in surface oceans has been largely documented (Roux et al., 2016a) are robust. Instead, we propose that long read assembly of VirION reads provides greater phylogenetic resolution of viral clusters by capturing members previously missed due to limitations in short-read assembly.

Figure 6 VirION-derived viral genomes from the Western English Channel are abundant in global marine viromes.

Relative abundances were calculated via competitive recruitment of 10 million sub-sampled reads from each of 42 samples from the Global Ocean Virome (Roux et al., 2016a). Short reads were recruited against a database comprising VirION-derived viral genomes (both scaffolded and un-scaffolded De Bruijn Graph (DBG) assemblies and those from long-read assembly of VirION long reads) and viral genomes obtained from other key viral metagenomic studies (including those which have employed short-read sequencing (‘GOV’; Roux et al., 2016a, and ‘Luo 2017’; Luo et al., 2017), and long-sequence recovery via Single-virus genomics (‘vSAG’; Martinez-Hernandez et al., 2017), and fosmid libraries (‘fosmid’; Mizuno et al., 2013; Mizuno et al., 2016), and viruses from the NCBI RefSeq database v. 8.4 (all detailed in Table S2). The Western English Channel sample is indicated with a ‘*’.

The most globally abundant and ubiquitous (identified in at least 10% of samples) viral genome was a contig from a hybrid assembly, denoted H_NODE_1248 (Fig. 7). This contig was 22.4 kbp in length and occupied a viral cluster (based on shared protein content) with 57 other members, including vSAG-37-F6 (9th most abundant ubiquitous virus and 13th most abundant across all samples), previously identified the most globally abundant virus (Martinez-Hernandez et al., 2017; Martinez-Hernandez et al., 2018). The viral cluster also contained 10 other contigs from long-read assembly of VirION reads, ranging in size from 10 kbp to 27 kbp. Interestingly, pelagiphage HTVC010P, once thought to be the most abundant virus on Earth (Zhao et al., 2013) was ranked 128th in global abundance and did not meet the criteria of being both ubiquitous (identified in at least 10% of the samples) and abundant (in the top 100 most abundant viral taxa for each sample). Upon its discovery as the most abundant global virus we previously urged a cautious interpretation as any representative of a new viral clade will recruit reads from all similar viruses in the environment (Zhao et al., 2013). As new representatives of these clades are captured in metagenomic data it is likely that competitive recruitment will split reads between all clade members, reducing the estimated abundance of any one single member.

Figure 7 Ubiquity of VirION-derived Western English Channel viruses in Global Ocean surface waters.

Heatmap shows the top 50 most abundant and ubiquitous (appear in >10% of samples) viral contigs in the surface samples of the Global Ocean Virome (Roux et al., 2016a). Competitive recruitment of 10 million subsampled short reads was performed using FastViromeExplorer (Tithi et al., 2018) against a contig database comprising: (1) viral population contigs from this study; (2) viral genomes derived from other key viral metagenomic studies (Table S2); (3) Viruses from the NCBI RefSeq database. Estimated abundances are calculated from the total number of reads mapped to a contig, with reads mapping to multiple contigs apportioned to a single contig via an expectation-maximum algorithm (Tithi et al., 2018). Matrix columns are ordered (left to right) by total number of mapped reads across all samples. The most abundant contig was H_NODE_1248, which is related at the genus level to the ubiquitous pelagiphage vSAG-37-F6. The Western English Channel sample is highlighted in a pink box, showing globally ubiquitous and abundant viruses from oceanic provinces were not particularly abundant in this coastal sample.

60% of the top 50 most abundant populations in the WEC were represented by a WEC contig derived from long-read assemblies of VirION reads (Fig. S7). The viral community in the WEC sample was dominated by a 39,972 bp circular genome from a hybrid assembly. Denoted H_NODE_525, this contig recruited 3.28 times more reads than the next most abundant contig (Fig. S7), but was not identified as globally abundant and ubiquitous (Fig. 7). This virus shared a viral cluster with the siphovirus Pseudoalteromonas phage vB_PspS-H6/1 but we were not able to determine its putative host despite using a variety of tools (Ahlgren et al., 2016; Galiez et al., 2017) (https://github.com/dutilh/CAT). A viral contig from hybrid assembly, denoted H_NODE_6 was the longest complete viral genome identified in this study, with a 316 kbp genome. In the short read-only assembly, this genome was broken into two contiguous contigs of 204 kbp and 112 kbp, respectively (Fig. S8). H_NODE_6 shared a viral cluster with the myoviruses Cronobacter sakasakii phage GAP32 and Enterobacter phage vB_KleM-RaK2. At 359 kb and 346 kb respectively (Šimoliūnas et al., 2012; Abbasifar et al., 2014), these are some of the largest phage genomes ever isolated. Recovery of this complete genome demonstrates the capacity for hybrid assembly with VirION reads to capture complete genomes of very large phages from complex communities on single contigs, which were fragmented using short-read only assemblies.

Conclusions

In summary, this investigation represents the first use of long-read sequencing for viral metagenomics. We have shown that using long-reads to scaffold short read De Bruijn Graph assemblies improves recovery of complete viral genomes. Furthermore, overlap-layout consensus assembly of VirION reads, followed by error correction with short reads captures abundant and ubiquitous viral populations that are missed (possibly as a result of genome fragmentation) by current short-read metagenomic methods. By combining these two approaches, our proposed bioinformatics pipeline maximises the capture of viral diversity whilst minimising the impact of high error rates associated with long-read sequencing and represents a major addition to the viral metagenomics toolset. Improved capture of viral genomic islands will enable better understanding of mechanisms underpinning host–virus interactions, as demonstrated in our capture of a shared genomic island on the newly observed HTVC010P-like pelagiphage tig404. Importantly, long-read sequencing on the MinION platform is undergoing rapid improvements in terms of yield, with current technology providing at least an order of magnitude more sequencing data than that produced in this study, at a cost of <$1000 per flowcell. Thus, our approach represents a significant advantage in terms of cost, yield and efficiency over fosmid and single-amplified genome approaches to capturing marine viruses that are otherwise challenging to assemble.

As error rates associated with MinION technology continue to fall, we envisage less and less complementary short-read data being required for polishing. A recent update to basecalling methods has led to a significant reduction in indel errors and their associated impact on protein prediction (Koren et al., 2019). Furthermore, there is no technical reason to prevent our VirION approach being used in conjunction with PacBio sequencing to further reduce error rates using circular consensus sequencing. Such an approach would remove the need for short-read error correction (Frank et al., 2016) and avoid the remaining indel errors observed following polishing of MinION read assemblies with short-read data. Reductions in DNA input requirements and/or improvements in DNA polymerases for increasing VirION amplicon lengths will further increase its utility in recovering viral genomes from metagenomic samples.

Ultimately, community efforts to align the input requirements of long-read sequencing with DNA recovery rates from viral communities will be rewarded by the ability to capture full-length viral genomes on single reads (Houldcroft, Beale & Breuer, 2017), including all associated nucleotide modifications (Viehweger et al., 2018). Oxford Nanopore sequencing interprets single stranded nucleotides as they pass through the pore and there are significant efforts to develop protocols for direct sequencing of RNA, dsDNA and ssDNA viruses (Keller et al., 2018; Viehweger et al., 2018; McCabe et al., 2018). It is theoretically possible that with the right combination of ligases and optimised buffers, we will soon be able to sequence dsDNA, ssDNA and RNA viruses, with associated nucleotide modifications, within a single library preparation. We expect that the VirION approach could be readily adapted for use with ssDNA or RNA viruses, provided appropriate amplification of starting material could be achieved to meet the requirements for sequencing. Therefore, the approach described here provides a significant step towards capturing the full diversity of the viral community.

VirION offers a framework for robust downstream bioinformatic approaches to maximise the benefits of long read sequencing, both now and as the technology continues to improve. Here, we have shown that VirION long-read metagenomics of dsDNA viral communities offers the potential to significantly improve our understanding of niche-differentiation, ecological speciation and the role of viruses in microbial communities within aquatic (Roux et al., 2016a) and soil (Pratama & Van Elsas, 2018) environments, human health (Mirzaei & Maurice, 2017; Aggarwala, Liang & Bushman, 2017) and industrial settings.

Supplemental Information

Supplementary Methods Supplementary Methods

Describes the quality control of sequence data and methods used to evaluate (1) relative abundances of contigs; (2) whether long read assembly captured more microdiverse genomes; (3) recovery of genomic islands and their predicted functional composition at the population level; (4) evaluation of functional variance of viral genomic islands spanned by VirION reads.

Click here for additional data file.

Table S1 Mock viral community member characteristics

Genomic characteristics of the six phages chosen for the mock viral community to develop and evaluate VirION protocols.

Click here for additional data file.

Table S2 The numbers of phage genomes identified in this study using short, hybrid and error-corrected long read assembly of VirION reads, as identified by VirSorter (Roux et al., 2015)

For comparison important viral metagenomic studies (see references) and viruses from ‘RefSeq’. Prior to quantification of global relative abundances and (shared-protein) clustering, phage genomes were re-analysed using VirSorter to ensure uniformity of gene-calling, resulting in above classifications. Note: VirSorter Categories as follows: 1 and 4: “most confident” predictions (viral and lysogen, respectively); 2 and 5: “likely” predictions (viral and lysogen, respectively).

Click here for additional data file.

Table S3 Student t-test results to identify significant differences between the number of circular viral contigs from short read only vs. hybrid assemblies

Student t-test results to identify significant differences between the number of circular viral contigs (as identified by VirSorter (Roux et al., 2015) from short read only vs. hybrid assemblies with VirION reads using metaSPAdes assemblies from triplicate random subsamples of short reads across different levels of sequencing depth. Significant differences are highlighted in bold.

Click here for additional data file.

Table S4 Student t-test results to identify significant differences between the number of viral contigs from short read only vs. hybrid assemblies with VirION reads

Student t-test results to identify significant differences between the number of viral contigs (as identified by VirSorter (Roux et al., 2015) from short read only vs. hybrid assemblies with VirION reads using metaSPAdes assemblies from triplicate random subsamples of short reads across different levels of sequencing coverage. Significant differences are highlighted in bold.

Click here for additional data file.

Table S5 Predicted genes located within 66 genomic islands spanned by VirION reads

For each spanning read, putative start and stop codons were estimated by hierarchical clustering and used as queries in a BLASTx alignment against the NR database. Genes with unknown function were removed and the remaining putatively classified genes were used to assess functional variance within viral genomic islands

Click here for additional data file.

Figure S1 Fragment length of LASL-amplified VirION reads before and after sequencing

(A) Bioanalyzer (Agilent) electropherogram showing the fragment length distribution of linker-amplified mock viral community DNA produced from 20 ng template DNA sheared to ∼8kbp. Amplicon length peaked at ∼5.4 Kbp, demonstrating PCR preference for amplification of shorter DNA fragments; (B) Read length distribution of VirION mock viral community amplicons (as shown in ‘A’; red dashed lines indicate approximate length of sheared template DNA); mean average read length was ∼4 kbp, likely due to preferential sequencing of shorter DNA fragments.

Click here for additional data file.

Figure S2 Evaluation of error correction of long-read assemblies using short read data

Impact of using short read sequencing to error correct overlap layout consensus-derived contigs with Pilon shows that approximate limits of the number of insertions and deletions that can be fixed is reached at ∼9 Gbp of short read data (median coverage of ∼70). Analysis was performed against the full contig set from Overlap layout consensus assembly of VirION reads from the Western English Channel ( n = 1,500).

Click here for additional data file.

Figure S3 Difference and 95% CI of median predicted protein length of different assembly types to evaluate the impact of sequencing error and error correction of VirION reads with short-read data

Median predicted protein length of 1,000 randomly selected proteins were calculated and compared to a similar treatment of proteins from a RefSeq v.8.4 Caudovirales database to measure effect size. This process was bootstrapped 1,000 times to provide 95% confidence intervals. The distributions on the graph represent distributions of differences in medians (Cumming, 2014) . The median effect size (bold number) and the 95% CI boundaries (black line under each distribution, and numbers in brackets) are shown.

Click here for additional data file.

Figure S4 Statistical significance of effects of assembly type on genomic island length and density

Effect size and bootstrapped median 95% CI intervals for impact of different assembly types on (A) genomic island length and (B) genomic island density (kbp of genomic island per kbp of genome). Values in boxes represent the median difference between 1,000 bootstrapped medians (95% CI). Green boxes represent significant ( p < 0.05) differences calculated with a Wilcoxon Rank Sum test.

Click here for additional data file.

Figure S5 Evaluation of genome-wide nucleotide diversity

The data point for long-read assembled contig tig404 (described in the main text) is highlighted; this virus belongs in the same viral cluster as pelagiphage HTVC010P, an abundant phage that fails to assemble in metagenomic datasets, potentially due to high microdiversity.

Click here for additional data file.

Figure S6 Alignment of the genome of HTVC010P with tig404 assembled using the VirION pipeline

Genomes were 89% identical at nucleotide in shared regions and both shared a conserved genomic island (green) bounded by structural proteins. Genome alignments were produced by Mauve (Darling et al., 2004) within the Geneious software (Kearse et al., 2012).

Click here for additional data file.

Figure S7 Top 50 most abundant viral contigs in a Western English Channel virome

Estimated relative abundances (number of recruited reads from the short-read dataset) of the Western English Channel viral contigs were calculated by competitive recruitment of short reads back to viral contigs derived from the VirION bioinformatics pipeline using FastViromeExplorer (Tithi et al., 2018). 60% of the top 50 most abundant viruses are detected only in the error-corrected overlap layout consensus assemblies.

Click here for additional data file.

Figure S8 The longest complete viral genome from our study was 316 kbp in length

H_NODE_6 was the longest recovered virus captured by scaffolding of a De Bruijn Graph assembly using VirION reads (red). Alignment of short read only contigs (blue) against the complete genome show the full length is only captured by the scaffolding approach, whereas the short-read approach results in a breakage at ∼205 kbp (grey box). Coverage and Shannon Entropy are both shown as median values of a 200 bp sliding window, with 100 bp overlap.

Click here for additional data file.

The authors thank the crew of the Plymouth Marine Laboratory vessel ‘Quest’ for collection of seawater samples, as well as Dr Simon Roux and Dr Benjamin Bolduc for guidance and advice on bioinformatic analyses. Portions of this research were conducted with high performance computing resources provided by Louisiana State University (http://www.hpc.lsu.edu), Ohio Supercomputer Center (1987), and the HPC infrastructure at University of Exeter.

Additional Information and Declarations

Competing Interests

Author Contributions

DNA Deposition

Data Availability

The authors declare there are no competing interests.

Joanna Warwick-Dugdale performed the experiments, analyzed the data, prepared figures and/or tables, authored or reviewed drafts of the paper, approved the final draft.

Natalie Solonenko performed the experiments.

Karen Moore performed the experiments, contributed reagents/materials/analysis tools.

Lauren Chittick performed the experiments.

Ann C. Gregory analyzed the data.

Michael J. Allen authored or reviewed drafts of the paper, approved the final draft.

Matthew B. Sullivan contributed reagents/materials/analysis tools, authored or reviewed drafts of the paper, approved the final draft.

Ben Temperton conceived and designed the experiments, analyzed the data, contributed reagents/materials/analysis tools, prepared figures and/or tables, authored or reviewed drafts of the paper, approved the final draft.

The following information was supplied regarding the deposition of DNA sequences:

Sequencing data and assemblies are available at the European Nucleotide Archive under the project accession number PRJEB27181.

The following information was supplied regarding data availability:

All code and analyses can be found in a GitHub repository: https://github.com/btemperton/long_read_viromics.

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
