# Peer review of "Long-read viral metagenomics captures abundant and microdiverse viral populations and their niche-defining genomic islands"

_PeerJ, doi:10.7717/peerj.6800_

## Round 0.1 · original submission · Minor Revisions

My apologies for the relatively length review process but I wanted to be certain that we got a number of unbiased reviews from multiple labs, and of course the holidays interfered. All of the reviewers were very complimentary of the manuscript and recommended a number of minor suggestions to improve it. Please attend to these points in a careful revision and I will endeavor to return a decision quickly. All the best.

Reviewer 1 ·

Basic reporting

no comment

Experimental design

no comment

Validity of the findings

no comment

Additional comments

The manuscript by Warwick-Dugdale et al describes a beautifully constructed study to evaluate an approach that will be transformative in viromics. The manuscript is extremely well-written and the conscientious attention to detail in this study is readily apparent.

Minor comments:
Line 114: should probably read “viral communities in seawater”

Line 476: Is the “and” supposed to be “or”?

Figure 7: Can the authors insert units for “Total abundance” and “Estimated abundance”? (in the figure or in the caption)

Reviewer 2 ·

Basic reporting

No comment. The article is very well put together, well written, and addresses all the goals that the study sought out to do.

Experimental design

The methodology produces novelty in the field of viromics and adds a workflow that will be useful for environmental virologists to further elucidate viral populations and communities. It would be worth mentioning something about ssDNA viruses and the ability to assess these members of the communities with these types of sequencing technologies (e.g. will the VirION workflow be able to identify these members in future work?). From my understanding this work only targets dsDNA members of the community. The research question was well-defined and addressed with the experiments and bioinformatic workflows. The methods described were extremely detailed and there is enough information to replicate the study. I would suggest thinking of ways to implement this pipeline as user-friendly for biologists, either through a GUI or some type of package. This is not necessarily required for this publication, just a suggestion to build this methodology so that other virologists using these sequencing platforms together in tandem would utilize this novel pipeline.

Validity of the findings

The novelty and results are very well demonstrated but some of the biological and ecological implications are touched on very lightly. Future directions for understanding how the VirION data can inform ecologists might be worth mentioning in the conclusions section in a little more detail (see lines 654-658). It also may be worth mentioning if this pipeline can be utilized with the sequencing of other environmental viruses that have different types of genomes, for example, RNA viruses and ssDNA viruses (i.e. this adds more breadth to the methodology for capturing the whole community).

Additional comments

I recommend to accept this article with minor revisions. My suggestion for minor revisions would be to address a few of the comments I have listed above in the "experimental design" and "validity of the findings" section. These are mostly just discussion points that could be addressed in the conclusion section.

Reviewer 3 ·

Basic reporting

This study has developed a technique to use both long and short read sequencing to overcome struggles in viral bioinformatics relating to difficulties in assembling viral genomes from metagenomes. I think it is timely and I appreciate the strategy of modifying or improving upstream aspects of sequencing and informatics as opposed to solely relying on assembly-based improvements.
Overall—great idea, addresses a known problem in viral ecology. Results seem promising in that it was successful in both mock communities and in the natural environment. My main criticism of this manuscript is that it is too long, and it is redundant in some places. I would suggest that some of the points in the Materials and Methods and Results are streamlined and that the repetition between sections is removed. Otherwise, I really appreciate this study, the novelty of the approach and the thoroughness with which the study was performed.

Experimental design

My only critique is that in some instances, the justification for using particular parameters is not well explained (eg. line 285-292). It is likely there are valid reasons for selecting particular parameters, however for broader audiences, it would be helpful for the justification, or citation to be listed.

Validity of the findings

Line 401 The indel rate is still rather high for the long read assemblies, which may bring into question the validity of this strategy for use with environmental samples. However, I still think this study is warranted and valid. The field cannot move forward with out these types of honest assessments of the current state of technology.

Additional comments

Line 95-97 Additionally, culture based approaches could improve the knowledge base.

Line 104-107 This sentence is confusing as worded.

Line 285-292 Why these particular parameters? There is no citation, and the number of reads, as well as the quality score selected are not justified.

Lines 354-360 This block of text repeats the methods.

Line 478, artifact not “artefact"

---

## Round 0.2 · accepted · Accept

It was a pleasure to have a manuscript reviewed so strongly by so many accomplished reviewers in the area of viral metagenomics.

#